# A protocol for recruiting and analyzing the disease-oriented Russian disc degeneration study (RuDDS) biobank for functional omics studies of lumbar disc degeneration

Olga N. Leonova[1]*, Elizaveta E. Elgaeva[2,3], Tatiana S. Golubeva[3], Alexey V. Peleganchuk[1], Aleksandr V. Krutko[4], Yurii S. Aulchenko[3☯], Yakov A. Tsepilov[2,3☯]*

1 Novosibirsk Research Institute of Traumatology and Orthopedics, Novosibirsk, Russia, 2 Novosibirsk State University, Novosibirsk, Russia, 3 Institute of Cytology and Genetics, Novosibirsk, Russia, 4 Priorov National Medical Research Center of Traumatology and Orthopedics, Moscow, Russia

☯ These authors contributed equally to this work.
* onleonova@gmail.com (ONL); tsepilov@bionet.nsc.ru (YAT)

## Abstract

Lumbar intervertebral disc degeneration (DD) disease is one of the main risk factors for low back pain and a leading cause of population absenteeism and disability worldwide. Despite a variety of biological studies, lumbar DD is not yet fully understood, partially because there are only few studies that use systematic and integrative approaches. This urges the need for studies that integrate different omics (including genomics and transcriptomics) measured on samples within a single cohort. This protocol describes a disease-oriented Russian disc degeneration study (RuDDS) biobank recruitment and analyses aimed to facilitate further omics studies of lumbar DD integrating genomic, transcriptomic and glycomic data. A total of 1,100 participants aged over 18 with available lumbar MRI scans, medical histories and biological material (whole blood, plasma and intervertebral disc tissue samples from surgically treated patients) will be enrolled during the three-year period from two Russian clinical centers. Whole blood, plasma and disc tissue specimens will be used for genotyping with genome-wide SNP-arrays, glycome profiling and RNA sequencing, respectively. Omics data will be further used for a genome-wide association study of lumbar DD with *in silico* functional annotation, analysis of plasma glycome and lumbar DD disease interactions and transcriptomic data analysis including an investigation of differential expression patterns associated with lumbar DD disease. Statistical tests applied in each of the analyses will meet the standard criteria specific to the attributed study field. In a long term, the results of the study will expand fundamental knowledge about lumbar DD development and contribute to the elaboration of novel personalized approaches for disease prediction and therapy. Additionally to the lumbar disc degeneration study, a RuDDS cohort could be used for other genetic studies, as it will have unique omics data.

**Trial registration number** NCT04600544.

**Funding:** The study is covered by the Russian Science Foundation grant number 22-15-20037. The data analysis will be performed using computational resources of the "Bioinformatics" Joint Computational Center supported by the budget project № FWNR-2022-0020. The funders had no role in study design, data collection and analysis, decision to publish, or preparation of the manuscript.

**Competing interests:** YSA is a founder and co-owner of PolyOmica and PolyKnomics, private organisations that provide services, research, and development in the field of quantitative and statistical genetics and computational genomics. This does not alter our adherence to PLOS ONE policies on sharing data and materials. Other authors declare no conflicts of interest.

## Introduction

Intervertebral disc degeneration is a normal aging process, but in some cases it causes lumbar disc degeneration disease (LDDD) [1,2]. Intervertebral disc degeneration (DD) often begins earlier than degenerative changes in the ligaments, cartilages and other tissues of spinal segment [3]. DD is a major contributor to low back pain [4], but can also proceed without back pain [5–8]. However, LDDD is associated with a lower health–related quality of life [9] and is the leading cause of population absenteeism and disability [10,11]. Intervertebral disc degeneration (DD) is an initiating factor in LDDD and is one of the major contributors to subsequent low back pain [4]. The prevalence of lumbar DD in the general population is extremely high: up to 50% in people aged 30–39 years [2,12]. There is not a single adult who has not at least some degree of degeneration in the intervertebral discs [13]. The list of well-known risk factors of LDDD includes female gender, advanced age [14], obesity, smoking [15], absence of or extreme physical activity [16] and genetic risk factors [17]. However, even in the absence of obvious risk factors, there are cases of LDDD among young people [1,14], as well as progressive severe LDDD requiring a number of surgical interventions.

The biology of intervertebral DD is extensively studied from different points of view. Besides various studies of degenerative disc morphology [18–22] and molecular biology [16,23–25], there is an increasing number of genetic [26,27], transcriptomic [28–30] and proteomic [31–33] studies of intervertebral DD. Thus, based on candidate-gene [34] and genome-wide associated studies (GWAS) [35,36], over 160 genes have now been considered to be potentially involved in intervertebral disc degeneration, although less than ten of them provide strong evidence for the association [34]. More than 500 genes have been highlighted as being expressed differently in degenerative disc tissue and healthy intervertebral discs [37,38]. Moreover, proteomic [31–33] and metabolomic [39] changes have also been detected in degenerative intervertebral discs. Despite a large number of LDDD studies, the exact mechanisms underlying this disease are not yet fully understood, partially because there are only few studies that use systematic and integrative approaches [40]. To our knowledge, there are no large-scale studies that integrate different genomics and other omics data within a single cohort. This complicates the extrapolation of previous findings on an integrative picture of LDDD pathology. Therefore, more integrative omics studies are needed to put all pieces of knowledge together, build a complete picture of the biology of lumbar DD and obtain a deeper understanding of the processes underlying this pathology.

Moreover, the low number of genes reliably and reproducibly associated with LDDD [34] does not explain its high heritability (up to 74%) [41]. It highlights that genetic and transcriptomic studies with greater sample sizes and replication samples are needed.

This protocol describes a study that provides broad opportunities for combining newly acquired and existing knowledge in various fields of genomics, transcriptomics and molecular biology due to its scale and the use of an integrative approach. Not only will this be beneficial for researchers as a resource of fundamentally new information on the mechanisms involved in disc degeneration pathology, it will enrich the list of reliable disease markers and drug targets, expanding the diagnostic and treatment approaches, which is crucial for the development of personalized, accurate and effective treatment of LDDD.

### Research aim and objectives

The main aim of this study is to establish a disease-oriented biobank to facilitate research into the biology of lumbar disc degeneration. Different biological samples (whole blood, plasma, disc tissue) along with MRI imaging, clinical, socio-demographic and various omics data (e.g. genomic and transcriptomic) will be available for researchers and clinicians for a variety of

further multi-omics studies. It will lay the groundwork for the development of early diagnostics of LDDD and its personalized treatment.

## Methods and analysis

### Study design and settings

This disease-oriented biobank to study lumbar disc degeneration will be recruited from two centers: the Priorov National Medical Research Center of Traumatology and Orthopedics (Priorov CITO), Moscow, Russia, and the Novosibirsk Research Institute of Traumatology and Orthopedics (NRITO), Novosibirsk, Russia. The study will be performed according to the Helsinki Declaration; the study protocol was approved by the Local Ethical Committee of the Novosibirsk Research Institute of Traumatology and Orthopedics (№034/20 dated 02 Oct 2020) and by the Local Ethical Committee of the Priorov National Medical Research Center of Traumatology and Orthopedics (№1/21 dated 25 Feb 2021). It will be conducted for a period of three years starting from 2021. Patients aged over 18 years with available MRI of the lumbar spine, who signed informed consent and met eligibility criteria, will be recruited. Clinical data and specimens will be collected in three visits as described in Table 1. At the first step, general information about a patient, his medical history and MRI scans of lumbar spine will be obtained. Patients' biological material (whole blood and plasma) will be sampled in the centers at baseline. Then, from those patients, who will undergo a spine surgery during the treatment disc tissue samples will be gained. Eventually, postoperative clinical data will be collected from operated patients during the follow-up. Over three years, we expect to collect information and biological material from a total of 1,100 patients. A full schematic of the study is presented in Fig 1.

All biological samples from NRITO will then be transported to the Institute of Cytology and Genetics (ICG), Novosibirsk, Russia, for long-term storage, further processing and omics

**Table 1. Scheduled procedures for clinical data and sample collection.**

|  | Visit 1—Baseline | Visit 2—Day of surgery[1] | Visit 3–3 months after surgery[2] |
|---|---|---|---|
| Visit Window (±Days) |  |  | Day 90 ±14 days |
| Informed Consent | X |  |  |
| Eligibility Criteria[3] | X |  |  |
| Medical History | X |  |  |
| Anamnesis Vitae | X |  |  |
| Demographics | X |  |  |
| Blood samples | X |  |  |
| MRI | X |  |  |
| ODI[4] | X |  | X |
| VAS[5] | X |  | X |
| Indications for surgery |  | X |  |
| Surgical Procedure |  | X |  |
| Intervertebral disc samples |  | X |  |
| Adverse events/Serious adverse events | X | X | X |

[1]only for patients subjected to surgery

[2]optional

[3]see Table 2 for details

[4]Oswestry Disability Index (ODI) scale

[5]Visual analogue scale (VAS) for back and leg pain.

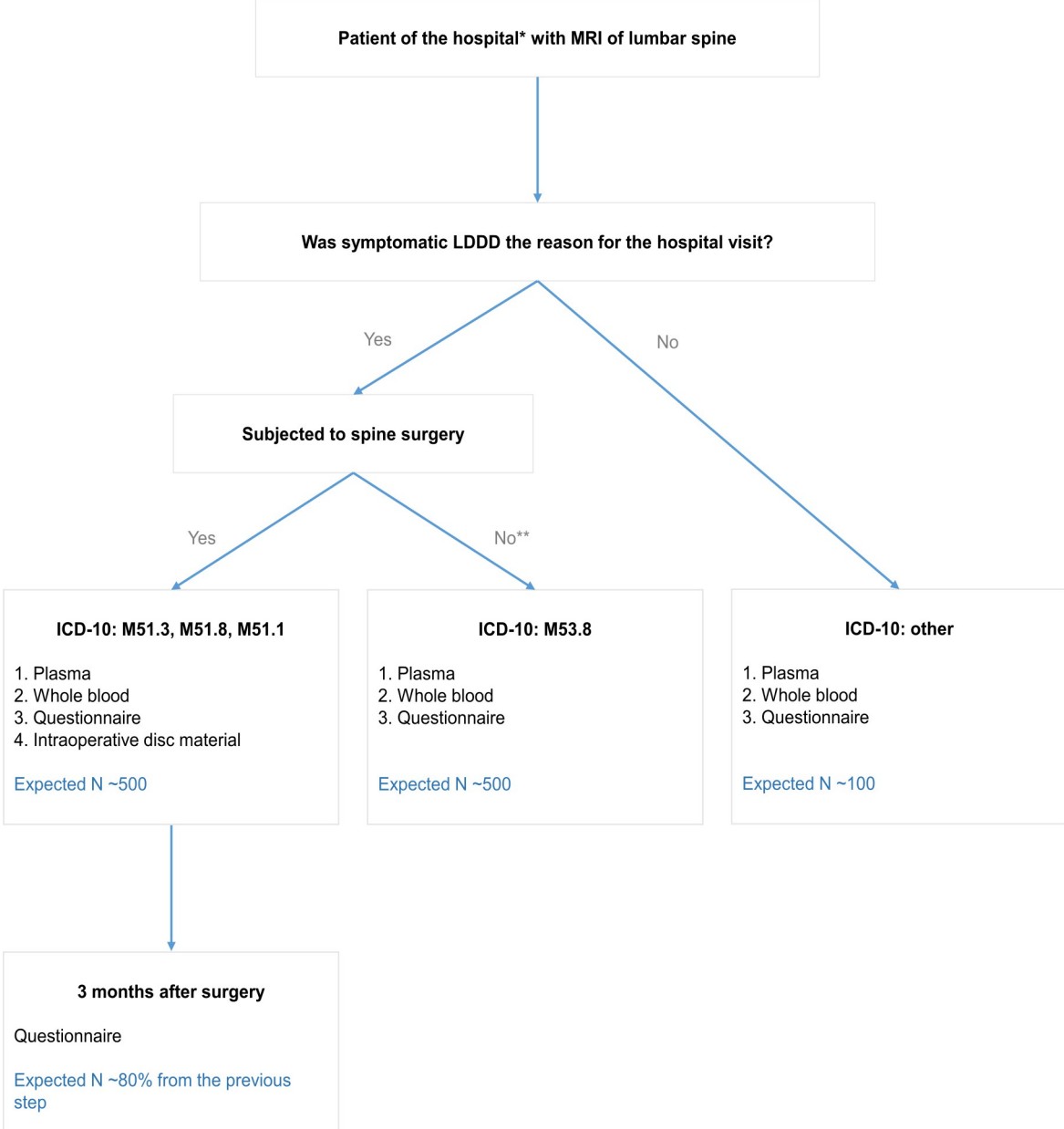

**Fig 1. Patient selection.** Sample size at various time-points of the project. Expected total sample size is 1,100. *Novosibirsk Research Institute of Traumatology and Orthopedics (NRITO) / Priorov National Medical Research Center of Traumatology and Orthopedics (Priorov CITO). **Conservative treatment.

profiling, including genotyping of blood samples and RNA sequencing of disc tissue. Priorov CITO will store the specimens on at their place.

## Patient selection and expected sample size

All patients of the two clinical centers aged over 18 with available MRI scans of the lumbar spine, who will sign an informed consent (see S1 File) and meet all the eligibility criteria (Table 2), will be included in the study. All participants will be assessed by the orthopedists and neurologists and grouped into three categories based on their diagnosis: patients with

Table 2. Eligibility criteria.

| Inclusion criteria | Exclusion criteria |
|---|---|
| 1. Age over 18;<br>2. Presence of lumbar MRI scans;<br>3. Signed informed consent for voluntary participation is provided. | 1. Any contraindication or inability to undergo baseline procedures;<br>2. Prior surgeries at any level of the lumbar spine;<br>3. Other non-degenerative spinal conditions that may have an impact on subject safety, wellbeing or the intent and conduction of the study;<br>4. History or presence of HIV, hepatitis B, hepatitis C. |

spine surgery, patients with conservative treatment of symptomatic LDDD and other patients with MRI scans of the lumbar spine (see Fig 1). If the reason for the hospital visit is symptomatic LDDD, patients will be divided according to the type of treatment (conservative or surgery) and corresponding ICD-10 codes. If the reason for the hospital visit is other than symptomatic LDDD, the patient will be included in the group of other patients with MRI scans of the lumbar spine in the study, regardless of the ICD-10 diagnosis code. Restrictions on participation in the study for patients with bloodborne pathogens are determined by the requirements of laboratory biosafety.

We plan to collect whole blood and plasma samples from not less than 1,100 patients (see Fig 1). The SNP array based whole-genome genotyping and total plasma protein N-glycosylation profiles will be measured for at least 384 participants. Total RNA sequencing expression profiles of approximately 40 disc specimens will be measured.

## MRI imaging, clinical and socio-demographic data

We require each patient from this study to have lumbar spine MRI images. MRI scanning will be performed on a 1.5 Tesla (or higher) tomograph and will include L1-S1 lumbar segments. T1 and T2 weighted images, acquired both in the axial and sagittal planes, will be obtained. Images will be used to assess the disc degeneration grade, presence and type of Modic changes and presence and severity of vertebral endplate defects. Disc degeneration grade will be estimated under the Pfirrmann classification from 1 to 5, with 1 corresponding to a normal disc and 5 corresponding to the most severe degeneration [42]. Modic changes will be evaluated for each endplate (from the lower L1 vertebra to the upper S1 vertebra) on sagittal scans using T1WI and T2WI [43]. Vertebral endplate defects will be ranged from 1 to 6 according to the Rajasekaran classification [44]. Then, total endplate scores (TEPS) will be calculated as the sum of the endplate defect scores of both upper and lower endplates in each L1-S1 spinal segment. Additionally, the height of the intervertebral discs and severity of osteophytes will be estimated based on Jarosz classification [45].

Clinical assessment will include demographic data (sex, age), self-reported ethnicity, height, weight, family status, physical activity, smoking, comorbidity, education level and job type.

All participants will complete the following clinical questionnaires: visual analogue scale (VAS) [46] back and VAS leg estimating the intensity of back and leg pain, correspondingly, and the Oswestry disability index (ODI) [47,48] questionnaire.

## Sample collection, transportation and storage

Samples collected in NRITO will be frozen at -40C (whole blood and plasma) or at -80C (disc samples) and then transferred to ICG for long-time storage at -80C. All biosamples from Priorov CITO will be frozen and stored on site at -80C.

Blood and plasma sampling: from each patient, 13 ml of peripheral venous blood will be collected into two BD Vacutainer 4-ml and 9-ml K2 EDTA tubes (or three 4-ml K2 EDTA tubes depending on availability). The tubes will be labeled with a patient's unique code. During

the first 30 min after blood collection, the 4-ml vacutainer with whole blood will be placed into a test tube rack and transported to the freezer to be stored at -40C. The 9-ml blood tube will be used for plasma extraction according to the plasma extraction protocol (see S2 File). Once the plasma has been extracted, the tubes with plasma will be labeled and placed into the freezer at -40C.

Eventually, all the tubes from NRITO will be transported to ICG in thermoboxes with cooling agents and then placed into freezers at -80C. Samples from Priorov CITO will be stored locally at -80C at once.

Collected plasma samples could be used for cell-free DNA, glycomics, proteomics and metabolomics analyses.

### Disc sampling

During the surgical procedure, the resected fragments of intervertebral discs will be placed into sterile falcon 50-ml tubes and stored in the operating room until the end of the intervention (for no longer than 40 min). The falcons will be labeled with unique codes and transported to the laboratory within 5 min where the discs' fragments will be put into liquid nitrogen.

Disc samples from NRITO will be transferred to ICG within thermoses with liquid nitrogen and then will be stored in freezers at -80C. Biospecimens obtained in Priorov CITO will be stored on site at -80C.

### Genotyping

DNA will be extracted according to the standard protocol using the Qiagen DNeasy Blood & Tissue Kit. Genotyping for not less than 600,000 SNPs will be performed using whole-genome SNP-arrays of high coverage (tentatively, Illumina Infinium Global Screening Array). Each genotyping batch will be sampled using the block randomization method implemented in the OSAT R package [49] by Bioconductor (functions "setup.sample", "setup.container", "create. experiment.setup" and "get.experiment.setup" with case / control LDDD status, sex, age, availability of the disc tissue, place of birth, BMI, physical activity level, smoking status and date of blood sampling used as characteristics for optimization). The imputation procedure will be carried out using the Haplotype Reference Consortium [48] or a later reference panel.

### Total plasma proteins N-glycosylation profiling

The plasma glycans profiling will be performed using the APTS glycan labelling kit (Genos) according to the standard protocol [49]. In short, plasma N-glycans are enzymatically released from proteins by PNGase F, fluorescently labeled with 2-aminobenzamide and cleaned up from the excess of reagents by hydrophilic interaction liquid chromatography solid phase extraction (HILIC-SPE) as described previously [50]. Fluorescently labeled and purified N-glycans are separated by HILIC on a Waters BEH Glycan chromatography column, $150 \times 2.1$ mm, 1.7 μm BEH particles, installed on an Acquity ultra-high-performance liquid chromatography (UHPLC) instrument (Waters, Milford, MA, USA) consisting of a quaternary solvent manager, a sample manager and a fluorescence detector set with excitation and emission wavelengths of 250 nm and 428 nm, respectively. Following chromatography conditions previously described in detail [50], glycan peaks (GPs)–quantitative measurements of glycan levels–are defined by manual integration of intensity peaks in the chromatograms. The abundance of N-glycans in each chromatographic peak is expressed as percentage area of the corresponding peak. Random allocation of the specimens will be carried out using the block randomization approach with the optimization characteristics similar to the genotyping procedure.

## Total RNA profiling

The intraoperative material will be homogenized using TissueLyzer II homogenator (QUA-GEN) and total RNA will be extracted and converted to cDNA using a kit for the isolation of total RNA and microRNA from cells and tissues (Biolabmix, Russia) and the M-MuLV–RH First Strand cDNA Synthesis Kit (Biolabmix, Russia), respectively. The amount of extracted RNA and its quality will be estimated using Bioanalyzer 2100 (Agilent).

Total RNA sequencing will be performed using Illumina-HiSeq 4000 under the PE-protocol. A read length of up to 100 bp and a sequencing coverage of 20M are expected. Specimens will be randomly selected for sequencing utilizing the block randomization approach in order to maintain a uniform distribution of optimization characteristics set for genotyping batches. In total, not less than 40 samples will be profiled.

## Duration of the project

Specimen collection and processing will last for three years starting in May 2021, unless additional funding has been obtained to expand the biobank. Samples will be stored for fifteen years after collection according to the storage protocol described above.

## Statistical data analysis

In this section, we will describe designs of further studies that we plan to conduct based on the omics data generated from the samples of the biobank.

**Genome-wide association study of lumbar DD and *in silico* follow-up.** We will perform a genome-wide association study (GWAS) using the GCTA software [51] using a mixed model linear regression model and quantitative scales of the assessed disc degeneration status. The GWAS results will be used for replication of genome-wide significant findings from the GWAS of DD conducted in 2013 by Williams et al. [35]. The associations reported by Williams et al. will be considered as replicated if the following two criteria are met: (1) the direction of the effect is the same in both studies; (2) the p-value of association in the replication sample is less than 0.05/4. Further, these two sets of GWAS will be meta-analyzed using the METAL software [52] to obtain the currently largest GWAS on lumbar DD for the European ancestry population. All loci passing the genome-wide significance threshold (p-value $< 5e-08$) for association will be identified using COJO [53]. The results of the meta-analysis will be used for *in silico* follow-up functional annotation and prioritization of genes involved in lumbar DD development.

**Sample size calculation and power estimation.** We do not set the upper limit for sample size, although we expect to recruit not less than 1,100 participants and perform a GWAS involving at least 384 patients. Thus, the total sample size of meta-analysis of the GWAS of lumbar DD conducted in 2013 by Williams et al. and in the current study will be about 5,000. This will allow us to detect the genomic loci with an odds ratio of 1.20 or higher with p-value $< 5e-8$ and 80% power [54].

**Functional analysis and data integration.** GWAS meta-analysis results will be annotated in order to predict the probable effects of replicated SNPs on gene expression and disease development. Furthermore, we will highlight the molecular pathways, cell and tissue types most likely to be involved in DD pathogenesis using tools like DEPICT [55] and FUMA [56]. This information will be used for gene prioritization alongside the results of the colocalization analysis [57] and Mendelian randomization [58,59]. This methodology will be applied for causal inference between DD and gene expression profiles from different tissues, also including the transcriptomic data from intervertebral discs obtained in this study. A gene network for LDDD regulation will be built and the key regulators will be revealed.

**Glycomics data analysis.** The association of plasma N-glycans levels with disc degeneration status will be studied. We are planning to analyze the differences in N-glycosylation patterns between patients with and without LDDD utilizing machine learning approaches including regression analysis. The results obtained could be helpful for the development of the dynamic glycan biomarkers for elucidating DD pathogenesis and for the development of a prognostic biomarker.

**Transcriptomic data analysis.** The patient group with available intraoperative disc material (see Fig 1) could be divided into "cases" or "degenerated disc" (grades 4–5 of disc degeneration according to the Pfirrmann classification) and "controls" or "healthy disc" (grades 1–3 of disc degeneration according to the Pfirrmann classification) [44,60].

This division does not influence the scheme of patient recruiting or sample collection but plays a role in the transcriptomic data analysis. The groups will include following ICD-10 codes: M51.1 "Thoracic, thoracolumbar and lumbosacral intervertebral disc disorders with radiculopathy" (cases); M51.3, M51.8 "Other thoracic, thoracolumbar and lumbosacral intervertebral disc degeneration" (controls).

Transcriptomic data obtained in this study will be used for detection of genes differentially expressed in discs between cases and controls. In short, the gene expression data will be mapped [61] onto the reference genome and the quality of the reads will be recalibrated ("Picard Toolkit." 2019. Broad Institute, GitHub Repository. http://broadinstitute.github.io/picard/; Broad Institute). Subsequently, aiming to reveal the transcriptomic differences in intervertebral discs tissue between cases and controls, we will count the reads and identify differentially expressed genes (DEGs) using the edgeR package [62]. To assess the statistical significance of the results, we will apply a Benjamini-Hochberg correction for multiple testing [63]. DEGs will be filtered by the q-value $< 0.05$ and divided into up- and down-regulated genes. Finally, the revealed DEGs will be functionally annotated with Gene Ontology terms.

The data will also be used to identify expression quantitative trait loci (eQTLs). We will perform an eQTL analysis of gene expression levels in intervertebral discs using the methodology described in [64]. Resulting regional association summary statistics will be used for gene prioritization.

## Data management

All data management and access procedures will be identical in both participating centers.

Each participant included in the study is assigned a unique code. Keys for these codes are saved in the locked storage of the internal hospital Electronic Data Capture (EDC) system, with access provided only for the curators of the study. All clinical data obtained from the patients are kept in the internal hospital EDC system accessible only by authorized researchers who are entering data into it. Information on who entered the data into the clinical database is available for viewing. During the study, internal monitoring will be conducted to maintain the quality of the study in accordance with the GCP principles. Participant and specimen codes are transferred to ICG along with the specimens from both clinical centers in an anonymous way.

Information on the physical location of specimens is kept with limited access only for curators of the study. Genotypes, transcriptomic data and data produced during their processing will be stored on a local server in ICG.

## Data and sample access

All clinically relevant data, such as genotypic data, medical history, MRI scans, results of transcriptome and N-glycome profiling could not be deposited in open access repositories due to

legal restrictions (data contains sensitive information). Access to this data and accesses to the biological samples could be granted on a collaborative basis upon special formal request to the data access committee of the Institute of Cytology and Genetics SB RAS. For further information, please contact data access committee (statgenomicslab@gmail.com).

The data generated during the specific analyses in the project will be made available to the scientific community following accepted standards, e.g. controlled access via such platforms as the European Genome-phenome Archive (EGA) (https://ega-archive.org/) for personally identifiable data, and public access for less sensitive summary-level data (e.g. GWAS results) via organization's web-site and/or such archiving services as Zenodo (https://zenodo.org/).

The study was registered at clinicaltrials.gov, trial registration number NCT04600544.

## Discussion

This study will be carried out in two different centers providing wider population coverage and more reliability to the sample storage, as data and specimens will be stored in two places. Participants will be recruited from unique medical centers (NRITO, Priorov CITO), which are the only centers in Russia specializing in DDD. The distinctive features of our clinical centers are a large constant flow of LDDD patients, the opportunity to conduct spine MRI and a highly-qualified team [65,66]. In the same way, our multidisciplinary research group, who will generate and analyze omics data, has a broad experience in studying back pain (one of the main manifestations of LDDD) [67–69] and statistical and functional data analyses using integrative omics approaches [70,71].

Another notable advantage of the project is the availability of diverse and rare biological material that can be used for multiple profiling. Besides whole blood samples, plasma and the intervertebral disc tissue obtained during spine surgery will be collected. The presence of different biological specimens for the same study participants will allow conducting integrative multi-omics analyses. Thus, not only genomes, but also transcriptomes, glycomes and other '-omes' can be measured using these samples. To our knowledge, this is the first study with a multiple omics profiling of similar biological material. Current studies are limited by a single profiling of one or two tissues: one in [32,33]; two in [28,29,31,37]; or use omics datasets in open access, combining data from different cohorts [40]. The importance of the multi-omics approach is hard to overestimate, as it allows us to look at the disease from different points of view and improve our understanding of the pathology.

One more substantial strength of our study is the assessment of the disc degeneration grade by MRI scans, because this is the most accurate and precise method of DD diagnostics. In LDDD studies based on plain radiography [72] or CT [73], phenotype definition is rather subjective as these approaches provide only indirect evidence of disc degeneration such as disc height loss and osteophytes. Similarly, the use of self-reported questionnaires for LDDD identification [74] is not reliable enough. By contrast, the Pfirrmann grading system of disc degeneration used in the present study is based on MRI scans and estimates the main characteristics of DD by assessing the signal intensity and the height of the intervertebral disc: the more dehydrated the disc is, the more severe degenerative changes are in it [42].

This study has some limitations. First, the expected sample size is modest in comparison with national country-level biobanks or large prospective cohorts. The sample size of the present study is one-fourth as large as the one reported for meta-analysis of LDDD studies by Williams et al. [35]. This limitation is explained by funding, establishing the recruitment period of three years. Nonetheless, the expected number of disc tissue samples (~510) and their transcriptome profiles (~40) is comparable with other omics studies using disc specimens

(proteome [33]: 7 cases / 7 controls; metabolome [39]: 60 cases / 21 controls; transcriptome [28]: 39 annulus fibrosus (AF) and 21 nucleus pulposus (NP) samples; [29]: 24 AF and 24 NP samples).

The next limitation of our study is an imbalance between patients with "healthy disc" and "degenerated disc" in the group of patients subjected to surgery. Obviously, patients with "healthy discs" are subjected to spinal surgery under specific circumstances. However, according to our protocol, we can still expect some controls in this group (in total, the expected number of controls is 20). In the groups not subjected to surgery, we expect a ratio between cases and controls to be closer to that in the general population.

It should be noted that "healthy disc" is a conditional definition, not objective enough; however, the inter-observer agreement is quite high among different groups of researchers, therefore, the Pfirrmann classification is considered to be highly reliable [75]. The main point here is that the existence of adult patients with perfectly healthy lumbar discs is questionable. To find such patients, a large-scale exploratory study is needed. Nevertheless, according to a population-based study, there are no adults with lumbar discs containing no degenerative changes [13].

Results of the work will be of practical importance, as they will replenish the theoretical base necessary for the development of a model of LDDD prediction, for example, polygenic risk score models based on genetic markers. Moreover, the data obtained could potentially make a valuable contribution to the development of new minimally invasive methods for degenerative spine condition treatment. Nowadays, molecular genetics and cell technologies are actively applied in practice to treat degenerative disc disease. In particular, increasing attention is being given to the development of molecular and gene therapy directed towards slowing and reversing disc degeneration [16]. It determines the social significance of this project.

In conclusion, this protocol describes the recruitment and analysis of the RuDDS biobank. The main goal of the project is to collate various medical data and biological specimens for further multidisciplinary investigations of LDDD. The notable advantages of the study are the MRI-based phenotyping and accessibility of a wide range of biological materials (MRI scans; blood, plasma and disc samples with omics data generated based on their analysis). The information gained from the collected biobank data will significantly contribute to the development of novel therapeutic approaches for LDDD treatment and qualitatively improve our understanding of the mechanisms underlying lumbar DD.

## Supporting information

**S1 File. Informed consent form for patients.**
(DOCX)

**S2 File. Standard operating procedure for plasma collection.**
(DOCX)

## Author Contributions

**Conceptualization:** Olga N. Leonova, Elizaveta E. Elgaeva, Aleksandr V. Krutko, Yurii S. Aulchenko, Yakov A. Tsepilov.

**Data curation:** Elizaveta E. Elgaeva, Aleksandr V. Krutko, Yakov A. Tsepilov.

**Formal analysis:** Elizaveta E. Elgaeva.

**Investigation:** Yurii S. Aulchenko.

**Methodology:** Olga N. Leonova, Elizaveta E. Elgaeva, Tatiana S. Golubeva, Aleksandr V. Krutko, Yakov A. Tsepilov.

**Project administration:** Olga N. Leonova, Aleksandr V. Krutko.

**Resources:** Alexey V. Peleganchuk, Yurii S. Aulchenko, Yakov A. Tsepilov.

**Writing – original draft:** Olga N. Leonova, Elizaveta E. Elgaeva, Tatiana S. Golubeva, Alexey V. Peleganchuk, Aleksandr V. Krutko, Yakov A. Tsepilov.

**Writing – review & editing:** Olga N. Leonova, Elizaveta E. Elgaeva, Tatiana S. Golubeva, Alexey V. Peleganchuk, Aleksandr V. Krutko, Yurii S. Aulchenko, Yakov A. Tsepilov.

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
