## [Decision Letter · Decision Letter 0]

6 Jan 2022

PONE-D-21-16554

Protocol for disease-oriented Russian disc degeneration study (RuDDS) biobank facilitating functional omics studies of lumbar disc degeneration

PLOS ONE

Dear Dr. Leonova,

Thank you for submitting your manuscript to PLOS ONE. After careful consideration, we feel that it has merit but does not fully meet PLOS ONE’s publication criteria as it currently stands. Therefore, we invite you to submit a revised version of the manuscript that addresses the points raised during the review process.

We look forward to receiving your revised manuscript.

Kind regards,

Walid Kamal Abdelbasset, Ph.D.

Academic Editor

PLOS ONE

Journal Requirements:

2. Thank you for stating the following in the Competing Interests section: "YSA is a co-owner of Maatschap PolyOmica and PolyKnomics BV, private organizations, providing services, research and development in the field of quantitative and computational genomics."

Reviewers' comments:

Reviewer's Responses to Questions

**Comments to the Author**

1. Does the manuscript provide a valid rationale for the proposed study, with clearly identified and justified research questions?

Reviewer #1: Yes

Reviewer #2: Yes

2. Is the protocol technically sound and planned in a manner that will lead to a meaningful outcome and allow testing the stated hypotheses?

Reviewer #1: Partly

Reviewer #2: Yes

3. Is the methodology feasible and described in sufficient detail to allow the work to be replicable?

Reviewer #1: Yes

Reviewer #2: Yes

4. Have the authors described where all data underlying the findings will be made available when the study is complete?

Reviewer #1: Yes

Reviewer #2: No

5. Is the manuscript presented in an intelligible fashion and written in standard English?

Reviewer #1: Yes

Reviewer #2: No

6. Review Comments to the Author

You may also provide optional suggestions and comments to authors that they might find helpful in planning their study.

Reviewer #1: Thank you for giving me this opportunity to review this article. The article is well written, though I have some serious concerns regarding the article.

Abstract:

1. Mention the study objectives in detail.

2. Include the allocation procedure in detail.

3. Mention the character of study participants.

4. Mention the statistical tests used for the study analysis.

Manuscript

1. Include the clinical significance of this study over clinicians, patients, and researchers.

2. Mention the eligibility criteria of the study participants.

3. Mention who has diagnosed the condition and is included in the trial?

4. Include the randomization and allocation procedure in detail.

5. Mention the method of sample size calculation with reference.

6. Mention the statistical tests used for the study analysis.

7. Summarize the contents of the discussion part.

Reviewer #2: Title

1. Title needs to be more specific

2. Kindly frame title such that it is accurate, informative, descriptive, succinct, simple and specific.

Abstract

1. Key words Better to be added at the end of the abstract

2. Methods section is poorly framed. It has to be re-written.

Introduction

1. English language need to be edited

2. Explain the rationale of the study ,Kindly focus on three elements of introduction.

a. What is known about the topic? (Background)

b. What is not known? (The research problem)

c. Why the study was done? (Justification)

Methods

1.Add references to methods section

2.Statistical methods that will be used t need to be explained in details

7. PLOS authors have the option to publish the peer review history of their article (what does this mean?). If published, this will include your full peer review and any attached files.

Reviewer #1: **Yes: **Gopal Nambi

Reviewer #2: No

---

## [Author Response · Author response to Decision Letter 0]

14 Mar 2022

Dear Editor,

We would like to thank you and the reviewers for all the helpful comments and suggestions. We have carefully revised the manuscript according to the comments provided by you and the reviewers and added the necessary clarifications and corrections. Please find our point-by-point responses below. We hope that you and the reviewers will find the revised manuscript suitable for publication in Plos ONE.

We include the updated conflict of interest statement: “YSA is a founder and co-owner of PolyOmica and PolyKnomics, private organisations that provide services, research, and development in the field of quantitative and statistical genetics and computational genomics. This does not alter our adherence to PLOS ONE policies on sharing data and materials. Other authors declare no conflicts of interest.” 

We would like to emphasize that this manuscript is a study protocol of biobank recruitment and analysis. Since the recruitment is an ongoing process, we do not report any data here. All clinically relevant data, such as genotypic data, medical history, MRI scans, results of transcriptome and N-glycome profiling could not be deposited in open access repositories. Access to this data and accesses to the biological samples could be granted on a collaborative basis upon special request to the data access committee of the Institute of Cytology and Genetics SB RAS. The summary-level data from the analyses (e.g. GWAS results) will be publically available via standard corresponding data sharing platforms (e.g. Zenodo, https://zenodo.org/). 

We have reformatted the “Data Access” section accordingly.

Yours Sincerely,

On behalf of the coauthors

Dr. Olga Leonova

Journal Requirements:

A.: We formatted the manuscript according to the requirements.

2. Thank you for stating the following in the Competing Interests section: "YSA is a co-owner of Maatschap PolyOmica and PolyKnomics BV, private organizations, providing services, research and development in the field of quantitative and computational genomics."

A.: We confirm it and added the sentence into Conflict of interest section.

A.: All clinically relevant data, such as genotypic data, medical history, MRI scans, results of transcriptome and N-glycome profiling could not be deposited in open access repositories. Access to this data and accesses to the biological samples could be granted on a collaborative basis upon special request to the Steering Committee of the Institute of Cytology and Genetics SB RAS. The summary-level data from the analyses (e.g. GWAS results) will be publically available via standard corresponding data sharing platforms (e.g. Zenodo, https://zenodo.org/). 

A.: Since it is a study protocol, we do not report any data here. The minimal sample size of the study is expected to be 1,100 patients with whole-genome genotyping and total plasma protein N-glycosylation profiles measured for at least 384 participants. Expression profiles of approximately 40 disc specimens will be measured.

Summary-level data generated based on omics profiles (e.g. GWAS results) will be made publically available after study completion via standard open sources for the corresponding data type sharing.

All clinically relevant data, such as genotypic data, medical history, MRI scans, results of transcriptome and N-glycome profiling could not be deposited in open access repositories due to legal restrictions (data contains sensitive information).

Individual-level data, such as medical history, MRI scans, genotypes, results of transcriptome and N-glycome profiling as long as biological specimens will be available upon special formal request to the data access committee of the Institute of Cytology and Genetics SB RAS. 

A:. Done.

Reviewers' comments:

Reviewer #1: Thank you for giving me this opportunity to review this article. The article is well written, though I have some serious concerns regarding the article.

Abstract:

1. Mention the study objectives in detail.

2. Include the allocation procedure in detail.

3. Mention the character of study participants.

4. Mention the statistical tests used for the study analysis.

A.: We rewrote the Abstract removing sectioning to make it fit the PLOS criteria. We have described the allocation details in Materials and Methods. 

Manuscript

1. Include the clinical significance of this study over clinicians, patients, and researchers.

A.: We added the clinical significance of this study over clinicians, patients, and researchers to the Introduction (see lines 75-80) and Discussion part (see lines 345-352).

2. Mention the eligibility criteria of the study participants.

A.: We added a more detailed description of participants of the study to the “Study design and settings” part of the manuscript (see lines 95-97). A detailed description of the eligibility criteria of the study participants had been given in the “Patient selection and expected sample size” section of the Methods part (see lines 119-129). The eligibility criteria are listed in Table 2.

3. Mention who has diagnosed the condition and is included in the trial?

A.: We added the description of who has diagnosed the condition of the study participants (see line 121; “Patient selection and expected sample size” section of the Methods part). More precise characteristics of the study participants were mentioned in the “Study design and settings” (see lines 95-97) and “Patient selection and expected sample size” (see lines 119-129) sections of the Methods.

4. Include the randomization and allocation procedure in detail.

A.: We included the randomization and allocation procedure description to the Methods part (see lines 179-183, 198-200, 208-210).

5. Mention the method of sample size calculation with reference.

A.: We added the description of sample size calculation to “Sample size calculation and power estimation” section of “Statistical data analysis” (lines 231-236).

6. Mention the statistical tests used for the study analysis.

A.: We added information on statistical tests to the Methods for further data analysis (see lines 223-228, 267-269).

7. Summarize the contents of the discussion part.

A.: We added a summary of the Discussion part (see lines 353-359).

Reviewer #2:

Title

1. Title needs to be more specific

2. Kindly frame title such that it is accurate, informative, descriptive, succinct, simple and specific.

A.: We corrected the title to make it more concrete. 

Abstract

1. Key words Better to be added at the end of the abstract.

A.: We added the key words at the end of the Abstract (see lines 39-40).

2. Methods section is poorly framed. It has to be re-written.

A.: We reformatted the Method section according to the comments of both Reviewers.

Introduction

1. English language need to be edited

A.: We have corrected English through the text.

2. Explain the rationale of the study ,Kindly focus on three elements of introduction.

a. What is known about the topic? (Background)

b. What is not known? (The research problem)

c. Why the study was done? (Justification)

A.: We have reformatted Introduction section according to the comments of both Reviewers.

Methods

1. Add references to methods section

2. Statistical methods that will be used need to be explained in details

A.: We have reformatted and updated the Methods section, added the information about statistical tests and methods (see the “Statistical data analysis”) and references.

---

## [Decision Letter · Decision Letter 1]

8 Apr 2022

A protocol for recruiting and analyzing the disease-oriented Russian disc degeneration study (RuDDS) biobank for functional omics studies of lumbar disc degeneration

PONE-D-21-16554R1

Dear Dr. Leonova,

We’re pleased to inform you that your manuscript has been judged scientifically suitable for publication and will be formally accepted for publication once it meets all outstanding technical requirements.

Kind regards,

Walid Kamal Abdelbasset, Ph.D.

Academic Editor

PLOS ONE

Additional Editor Comments (optional):

Reviewers' comments:

Reviewer's Responses to Questions

**Comments to the Author**

1. Does the manuscript provide a valid rationale for the proposed study, with clearly identified and justified research questions?

Reviewer #2: Yes

2. Is the protocol technically sound and planned in a manner that will lead to a meaningful outcome and allow testing the stated hypotheses?

Reviewer #2: Yes

3. Is the methodology feasible and described in sufficient detail to allow the work to be replicable?

Reviewer #2: Yes

4. Have the authors described where all data underlying the findings will be made available when the study is complete?

Reviewer #2: Yes

5. Is the manuscript presented in an intelligible fashion and written in standard English?

Reviewer #2: Yes

6. Review Comments to the Author

You may also provide optional suggestions and comments to authors that they might find helpful in planning their study.

Reviewer #2: Thanks for submitting the required modification and I hope you finish your work properly as described and mentioned in your study protocol

7. PLOS authors have the option to publish the peer review history of their article (what does this mean?). If published, this will include your full peer review and any attached files.

Reviewer #2: **Yes: **Marwa Eid

---

## [Editor Report · Acceptance letter]

5 May 2022

PONE-D-21-16554R1 

A protocol for recruiting and analyzing the disease-oriented Russian disc degeneration study (RuDDS) biobank for functional omics studies of lumbar disc degeneration 

Dear Dr. Leonova:

I'm pleased to inform you that your manuscript has been deemed suitable for publication in PLOS ONE. Congratulations! Your manuscript is now with our production department. 

Kind regards, 

on behalf of

Dr. Walid Kamal Abdelbasset 

Academic Editor

PLOS ONE